# Unlocking ultrafast hot hole transport in transition metal oxides governed by the nature of optical transitions

Keming Li[1,8], Yingjie Wang[1,8], Lan Jiang [1,2,3], Guoquan Gao[1], Guanzhao Wen[4], Yan Zhang[5], Xianjie Wang[6], Shuaifeng Lou [5], Mischa Bonn [4], Hai I. Wang [4,7] ✉ & Tong Zhu[1] ✉

The intrinsically low carrier mobility of transition metal oxides within the polaron transport framework fundamentally limits their optoelectronic performance. Although optical transitions profoundly impact carrier generation and transport dynamics in oxide systems, the underlying mechanisms remain elusive. Here we demonstrate that the nature of optical transitions decisively regulates hot-hole transport in representative oxides, $Co_3O_4$ and $\alpha$-$Fe_2O_3$. Combining ultrafast optical nanoscopy with terahertz spectroscopy, we identify two distinct regimes: rapid band-like transport of energetic holes within a few picoseconds (~100 $cm^2 s^{-1}$) and slower polaron-dominated hopping transport (~$10^{-3} cm^2 s^{-1}$) thereafter. Both the oxide composition and the transition pathway play critical roles in tailoring sub-picosecond hot-carrier dynamics. In $Co_3O_4$, metal-to-metal excitation at 1.55 eV yields an ultrahigh diffusion constant of 290 $cm^2 s^{-1}$, seven times that generated by higher-energy ligand-to-metal transitions (2.58 eV). These findings underscore the pivotal role of transient hot-carrier dynamics and suggest optical control of excited states as a promising route for optimizing energy management in oxide-based optoelectronic and photocatalytic systems.

Transition metal oxides (TMOs) find widespread integration into photocatalysis[1–3], photovoltaics[4–6], and photodetection[7] systems, thanks to their stability, eco-friendly nature, and suitable band gaps[8,9]. One of the key challenges in such photoelectrochemical (PEC) systems lies in minimizing photocarrier energy loss in fulfilling the intended functions[3,10,11]. Efficient transport of charge carriers can reduce recombination and enhance the separation efficiency of electrons and holes, a prerequisite for the efficient operation of PEC systems. Most TMOs demonstrate slow, thermally activated hopping transport dominated by polaronic states. Polaron formation—charge carriers dressed with a local lattice deformation—is ubiquitous in

TMOs due to their strong ionic character and resultant strong interactions between photocarriers and the lattice[12–14]. This leads to a transport bottleneck with a diffusion length of only a few nanometers, as described by the Gartner model[15,16]. However, recent studies have reported unexpectedly long-range transport of photo-generated holes in TMOs[17]. Notably, under backside illumination, the energetic holes generated under high photon energy excitation (with shorter penetration depth, and excitation occurring farther from the depletion region) surprisingly yield higher photocurrent compared to near-band-edge excitation[17]. These findings directly challenge the framework of inefficient ground-state polaron transport and suggest

[1]Laser Micro/Nano Fabrication Laboratory, School of Mechanical Engineering, Beijing Institute of Technology, Beijing, China. [2]Beijing Institute of Technology Chongqing Innovation Center, Chongqing, China. [3]Yangtze Delta Region Academy, Beijing Institute of Technology, Jiaxing, China. [4]Max Planck Institute for Polymer Research, Mainz, Germany. [5]State Key Laboratory of Space Power-Sources, Harbin Institute of Technology, Harbin, China. [6]School of Physics, Harbin Institute of Technology, Harbin, China. [7]Nanophotonics, Debye Institute for Nanomaterials Science, Utrecht University, Utrecht, the Netherlands. [8]These authors contributed equally: Keming Li, Yingjie Wang. ✉e-mail: h.wang5@uu.nl; tongzhubit@bit.edu.cn

that photon energy may have a broader impact on carrier dynamics in TMOs than previously anticipated.

Extensive incident photon-to-current efficiency (IPCE) measurements at the device level have also provided valuable insights into charge carrier generation, transport, and collection in TMO photoanodes. For instance, in hematite ($\alpha$-Fe$_2$O$_3$), Huang et al. reported that the IPCE spectrum deviates significantly from the absorption spectrum, particularly exhibiting a decline in internal quantum efficiency near the bandgap[18]. This observation is in drastic contrast to conventional solar cell materials such as silicon, where IPCE is independent of the excitation wavelengths. Recent studies attribute this efficiency dependence of excitation wavelength in TMOs to the nature of optical transitions[19,20]. Under the combined effects of Mott-Hubbard splitting and crystal field splitting, electronic transitions in TMOs can be categorized into ligand-to-metal charge transfer (LMCT) and metal-to-metal transitions (MMT) originating within or between metal d orbitals[21]. Grave et al. reported a positive correlation between the carrier mobility and charge generation quantum yield product ($\mu\phi$) to the IPCE, demonstrating the nature of optical transitions underlying the photocurrent generation in TMOs[22]. This correlation is further supported by the observed wavelength-dependent relationship between quantum yield, LMCT bands, and photocurrent spectrum, which indicates that the transport capacity of the carriers determines the upper limit of the achievable photocurrent[18,23,24]. However, the mechanism by which the optical transition type affects the carrier transport and the quantum yield remains poorly understood. Carneiro et al. suggested that the non-thermal phonon bath under high-energy excitation increases the hopping radius and mobility of ground-state polarons[25]. While this explanation provides insight into carrier transport under specific conditions, the thermally activated hopping transport by polarons does not account for the observed wavelength-dependent IPCE. Furthermore, high-energy excitation primarily impacts non-equilibrium dynamics, where carriers may exhibit transient behaviors that deviate from steady-state assumptions. This highlights the need for in-depth investigating non-equilibrium dynamics to elucidate how different optical transitions modulate carrier behavior through their transient effects and interplay with excitation conditions. One of the challenges lies in directly probing the charge transport effects in TMOs in real-time with sub-ps time resolution and real-space with nanometer spatial accuracy.

Here, by combining ultrafast optical nanoscopy with terahertz (THz) spectroscopy, we reveal efficient transport of hot holes in two representative TMOs, Co$_3$O$_4$ and $\alpha$-Fe$_2$O$_3$, closely related to the types of optical transitions. In Co$_3$O$_4$, optical excitations to LMCT by 2.58 eV result in hot holes with a high diffusion constant of 41 cm$^2$ s$^{-1}$ within the first ps. Remarkably, the hot hole diffusion constant is effectively modulated by tuning the excitation photon energy h$\upsilon$: in contrast to the conventional expectation that higher h$\upsilon$ leads to more efficient hot carrier diffusion, our result demonstrates that the nature of the charge state that is optically accessed plays a fundamentally important role in tailoring sub-ps hot carrier transport dynamics. Excitations to MMT states by 1.55 eV give rise to delocalized hot holes which can diffuse over 200 nm within 1 ps, i.e., with a diffusion constant approaching 300 cm$^2$ s$^{-1}$. After 1 ps, polaron-dominated hopping transport significantly reduces carrier diffusion constant down to 5*10$^{-3}$ cm$^2$ s$^{-1}$, five orders of magnitude lower than that of transient hot states due to strong carrier-phonon interactions. THz spectroscopy further confirms the delocalized nature of sub-ps hot carriers via temperature-dependent photoconductivity studies. Moreover, aiming to provide a complete picture of ultrafast hot hole transport in TMOs, we extended our investigations to another open d-shell TMO, $\alpha$-Fe$_2$O$_3$. We observe hot hole diffusion exceeding 450 nm within ~2 ps by pumping LMCT transitions. Overall, our results provide a new perspective on carrier transport in TMOs, with the ultrafast transport of hot holes exhibiting transport speeds up to nearly five orders of magnitude higher than

previously reported steady-state values. These findings reveal the efficient hot carrier transport of TMOs and its potential contribution to device performance, providing an innovative insight into energy management of TMO-based PEC systems.

## Results

Co$_3$O$_4$ was deposited on a quartz substrate by pulsed laser deposition. Raman Spectroscopy results (Supplementary Note 1) confirmed its high quality and crystallinity. Co$_3$O$_4$ has a spinel structure in which cobalt exists in two oxidation states, Co$^{2+}$ and Co$^{3+}$, occupying tetrahedral and octahedral symmetry positions with a stoichiometric ratio of 1:2, respectively. Owing to the combined effects of Mott-Hubbard and crystal field splitting, the 3d orbitals of Co divide into e$_g$ and t$_{2g}$ orbitals. The conduction band of Co$_3$O$_4$ is contributed by the 3d orbitals of both Co$^{2+}$ and Co$^{3+}$, while the valence band is contributed by the 2p orbitals of oxygen and the 3d orbitals of metal cations[26]. As shown in Fig. 1a, the steady-state absorption spectrum of the Co$_3$O$_4$ film exhibits several distinct absorption peaks (2.9 eV, 1.64 eV, 0.98 eV, and 0.84 eV). The 2.9 eV absorption peak is attributable to LMCT between oxygen and Co$^{2+}$, while MMT within or between Co$^{2+}$ and Co$^{3+}$ accounts for the remaining transitions[27,28]. Detailed parameters for Gaussian fitting of the absorption spectrum can be found in Supplementary Table 1. Employing the Tauc method, the band gap of Co$_3$O$_4$ is estimated at 0.790 ± 0.001 eV (Supplementary Fig. 3), which is consistent with previous reports[28–30].

### Assigning optical transitions in Co$_3$O$_4$

To investigate the dynamics of photocarriers in Co$_3$O$_4$, transient absorption (TA) spectroscopy measurements were systematically performed. In the study, a femtosecond pump laser pulse excites valence band electrons, while a supercontinuum white light pulse is used to track the dynamics of transient absorption changes due to carriers populating the sample. We present the time-dependent TA spectrum, i.e., differential absorption spectrum with/without excitations by 700 nm (equivalently 1.77 eV) in Fig. 1b. Negative signals, e.g., at 435 nm and 750 nm, are related to photoinduced bleach (PIB), corresponding to two absorption peaks in the steady-state absorption spectrum. The broad positive signal from 800 nm to 1200 nm is caused by the photoinduced absorption (PIA), which can be assigned to intraband electronic excitations in the conduction band[31]. Strong electron-phonon interactions lead to the localization of these electrons and the formation of small electron polarons, resulting in long-lived PIA at 910 nm, consistent with the findings of Zhang et al.[31]. Importantly, the dynamics of pump-induced changes at 435 nm (related to PIB) and 910 nm (PIA) demonstrate nearly identical decay constants as shown in Fig. 1c, indicating the same type of charge carriers involved for these two transitions. As the PIA signal at 910 nm is known to relate to electronic excitations, this result suggests that the optical transition at 435 nm originates from interband excitations involving the conduction band edge electrons.

The dynamics at 600 nm, on the other hand, exhibit a distinct decay profile compared to other wavelengths (Fig. 1c). To further identify the origin of this signal, we performed TA measurements on Co$_3$O$_4$ in the presence of selective electron and hole scavengers (Supplementary Note 3). In the presence of a hole scavenger (MeOH), the 600 nm signal decays significantly faster, indicating that it originates from photoinduced hole absorption. By contrast, its kinetics remain unchanged in the presence of an electron scavenger (AgNO$_3$), further corroborating this assignment. This interpretation is consistent with previous spectroelectrochemical studies and with the band structure calculations of Co$_3$O$_4$[28,29]. We therefore reasonably attribute the unique PIA at 600 nm to photoinduced hole absorption arising from optical transitions from O 2p orbitals to the valence band maximum. It is noteworthy that the PIA signal blueshifts within ~1 ps (Fig. 1d). This shift cannot be attributed to Coulomb-induced bandgap

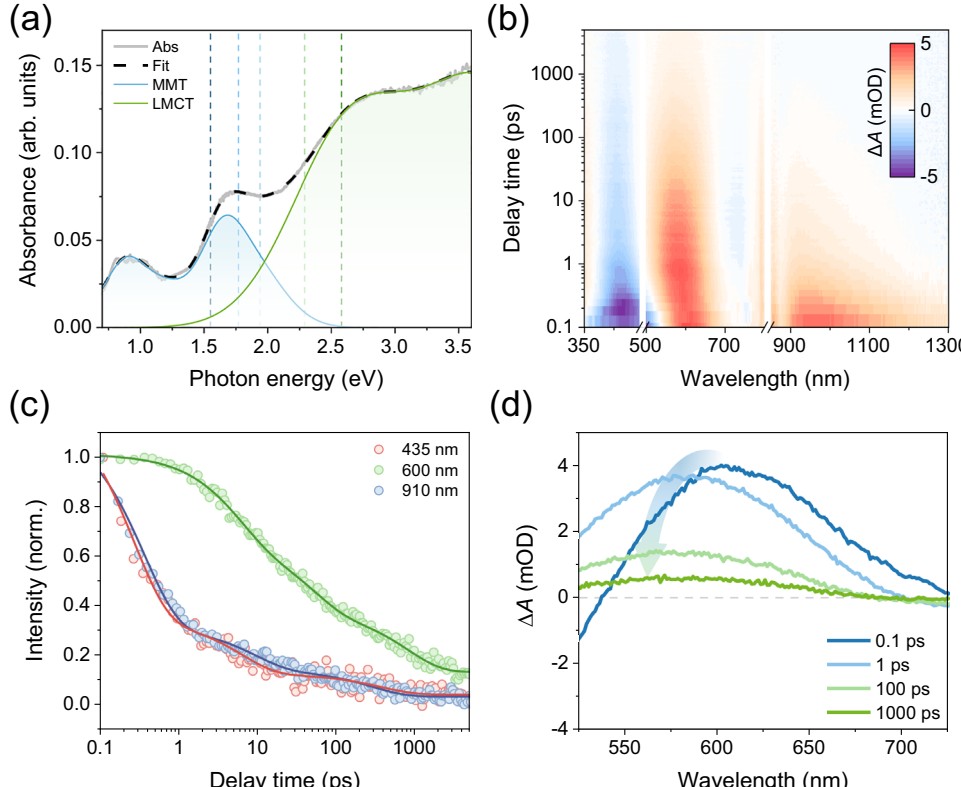

**Fig. 1 | Co₃O₄ transient carrier dynamics. a** Steady-state absorption spectrum of the $Co_3O_4$ film. The spectrum was fitted (black dash line) to the sum of six Gaussian bands following previous reports[28]. The absorption bands assigned to MMT and LMCT transitions are shown in blue and green, respectively. The vertical dashed lines indicate the pump photon energies used in the ultrafast optical nanoscopy measurements. **b** 2D color plot of TA spectra of $Co_3O_4$ film at 1.77 eV pump photon energy (**c**) TA dynamics at 910 nm, 600 nm, and 435 nm. **d** Representative TA spectra of the $Co_3O_4$ at indicated delay times. Arrows are used to guide the eye and indicate the spectral shift.

renormalization, which typically results in carrier density-dependent shifts and is accompanied by bandgap recovery[32]: as shown in Supplementary Fig. 4, our TA results revealed no such power dependence, nor did we observe further spectral changes within the examined time window of up to 5000 ps. Additionally, defect trapping can also be ruled out due to the lack of variation in decay rates across different excitation fluences. This is because trap filling effects have been previously shown to modulate charge carrier dynamics by tuning excitation densities[33]. Instead, the formation of small hole polarons, driven by strong carrier-phonon interactions inherent to TMOs could rationalize the observed dynamics, as discussed in Supplementary Note 4[34–38]. The nature of this coupling is particularly prominent in carrier cooling and transport, which significantly affects the fate of carriers. Subsequent transport measurements further support the formation of small hole polarons in the longer (~1 ps) time scale.

### Real-time real-space tracking of hot carrier diffusion in Co₃O₄ by ultrafast optical nanoscopy

Next, we employed ultrafast optical nanoscopy to investigate the carrier transport properties of $Co_3O_4$. To visualize the spatiotemporal evolution of specific photocarrier species, the pump beam is held at a fixed position while the probe beam is scanned relative to the pump with a two-dimensional (2D) galvo mirror. The pump-induced change in probe transmission $\triangle T$ is collected and mapped as a function of probe position to form an image (Fig. 2a, See Methods for more information)[39–41]. Here we focus on probing the hole dynamics as hole transport plays a key role in determining the performance of photoanodes based on TMOs[3]. To this end, we first excite the MMT mode by 700 nm (1.77 eV) pulses and then monitor the spatiotemporal dynamics of photogenerated holes by probing the associated

transition at ~570 nm. Quantification of the holes' spatial distribution is performed by fitting the data to a 2D Gaussian function (Fig. 2b), where the variance $\sigma^2$ represents the extent of hole spatial distribution. The diffusion of holes away from the initial excitation volume is reflected by an increase in mean-squared displacement (MSD = $\sigma_t^2 - \sigma_0^2$, where $t$ is the delay time, and $\sigma_t^2$ and $\sigma_0^2$ are the spatial variances at the delay time $t$ and 0, respectively). The evolution of MSD as a function of pump-probe delay time is illustrated in Fig. 2c. The hole transport demonstrates two distinct regimes and the diffusion constant $D$ can be determined using $D = \frac{\sigma_t^2 - \sigma_0^2}{2t} = \frac{MSD}{2t}$. Two distinctive features are observed in the inferred time-dependent diffusion constant (independent of film thickness, see Supplementary Note 6): First, in the longer time scale, e.g., for the pump-probe delay beyond 1 ps, charge carriers diffuse slowly with a constant of merely 0.005 ± 0.003 cm² s⁻¹, which is consistent with polaron formation[42]. Once small polarons are formed by strong carrier-phonon interaction, their increased effective mass and self-trapping effects lead to a significant decrease in the carrier transport capacity[14,42]. The polaron formation time scale (~1 ps) is also in line with a blue shift observed in the PIA signal (see detailed discussion in Supplementary Note 7) and THz photoconductivity studies discussed in the next section. Second, in the sub-ps timescale, we observe a remarkably fast diffusion of non-equilibrium hot holes with a rate up to over 150 cm² s⁻¹ following MMT excitation. This result indicates the transient delocalized nature of charge carriers before they condensate into polarons, and may strongly impact on the charge carrier generation and capture in photochemical cells.

To further unveil how optical transitions impact the transient ~ ps photocarrier transport in TMOs, we conducted a series of ultrafast optical nanoscopy measurements on $Co_3O_4$ thin films using various

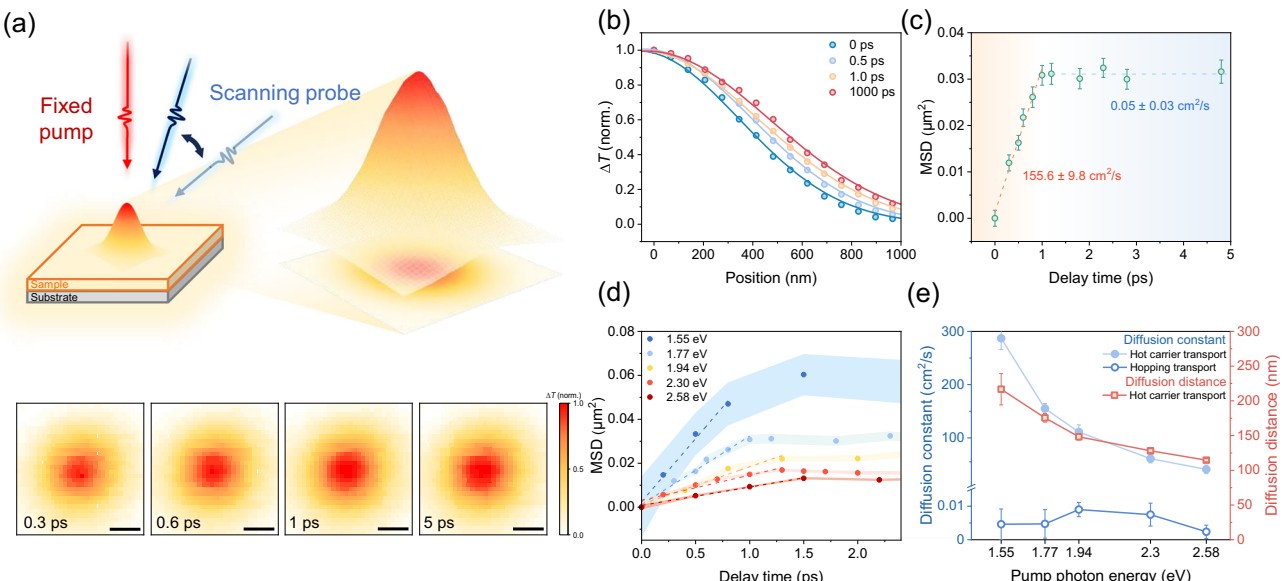

**Fig. 2 | Spatially resolved dynamics of photogenerated holes in $Co_3O_4$.**
**a** Working principle and data of ultrafast optical nanoscopy. In the upper image, the distribution of photogenerated holes can be satisfactorily described using a 2D Gaussian fit. The lower images present 2D mapping of ultrafast optical nanoscopy images of $Co_3O_4$ under a pump photon energy of 1.77 eV at various time delays, showing a rapid expansion of the hole distribution within 1 ps. Scale bars, 500 nm. **b** 2D Gaussian cross section in the X direction with representative delay times. The curves represent Gaussian fits. **c** Time evolution of MSD extracted from 2D Gaussian fitting. Linear fits are shown in dashed lines to indicate diffusive transport. The corresponding diffusion constant is included. Error bars represent 2D Gaussian fitting errors. **d** Evolution of the MSD at different pump photon energies. The error bar is represented by color shading. The dashed line represents a linear fit for the initial diffusive transport. Error bars represent Gaussian fitting errors. **e** Hole diffusion constant of two transport regimes and diffusion length of hot hole transport regime as a function of pump photon energy. Error bars of diffusion constant represent linear fitting errors; diffusion length errors are propagated from the diffusion constant.

pump photon energies hυ while maintaining a constant photocarrier density. It should be noted that, due to the spectral overlap among different optical transitions, excitation at a given photon energy can simultaneously activate multiple excitation pathways. As a result, the measured diffusion constant reflects a population-averaged behavior of multiple hole species. In this study, we focus on qualitatively correlating the observed diffusion trends with the dominant excitation characteristics under different pump wavelengths. We tune the pump photon energy from 1.55 eV to 2.58 eV, by which the optical transition proportion in $Co_3O_4$ shifted gradually from MMT-dominated to LMCT-dominated. Firstly, for all excitations, we found little spatial expansion of charge carriers after ~1 ps (Fig. 2d and Supplementary Note 8), in line with our assignment of polarons being the final equilibrium states independent of the optical transitions. Second, delocalized charge species with high diffusion constants in the sub-ps time scale are observed for all excitation wavelengths; yet, the carrier diffusivity depends strongly on the pump wavelength. The hot hole diffusion constant is found to decrease from $287 \pm 21 \, cm^2 \, s^{-1}$ for the MMT-dominated transition at 1.55 eV to $41 \pm 8 \, cm^2 \, s^{-1}$ for the LMCT-dominated transition at 2.68 eV (Fig. 2e), despite a large increase in the pumping hυ in the latter case.

The comparison of the two transport regimes, i.e., sub-ps rapid hot carrier diffusion vs slow polaron conduction for >2 ps time scale, further highlights the significant yet previously underestimated role of hot hole transport. Furthermore, we note that previous spatial-temporal studies employing the same tool have unveiled fast diffusion of hot carriers in perovskite materials on a similar sub-ps time scale[39,43]. There, the diffusivity of hot carriers increases with the excess energy (defined as the energy difference between hυ and optical bandgap $E_g$). Our results unveil the particularity of the hot carrier diffusion in $Co_3O_4$: The nature of optical transitions, rather than the excess energy, dictates the hot carrier diffusivity. Hot hole states (related to oxygen orbitals) generated by LMCT, despite possessing much higher excess energy, appear much less delocalized than those of the hole states associated with cobalt orbitals followed by MMT[44].

While our study unveils the orbital-dependent, highly diffusive nature of transient hot carriers in $Co_3O_4$, one critical question lies in the extent to which this effect impacts charge carrier collection in practical applications, given the short-lived nature of these states. To address this, we further infer the total diffusion length ($L$) of holes, and the sub-ps contribution by hot carriers under 1.55 eV excitation. Despite its short lifetime, hot carriers can contribute substantially to the overall carrier diffusion in $Co_3O_4$: while we show that hot holes generated by 1.55 eV excitation diffuse over 200 nm within 1 ps, the "cold" hole polaron, with a lifetime of approximately 800 ps, only reaches a diffusion length of about 30 nm. This suggests that hot carriers account for about 90 % of the total diffusion length of delocalized carriers, highlighting their crucial role in transport kinetics (see more details in Supplementary Note 9). The total hole diffusion length in this work is nearly two orders of magnitude larger than previously reported values for similar TMOs[15]. Even under 2.58 eV excitations to the LMCT transition with less delocalized holes, the hot hole diffusion constant still exceeds the experimental values obtained through electrical measurements by five orders of magnitude[45,46]. This huge discrepancy in measured transport rate and diffusion length is likely due to the lack of time resolution in static transport measurements. Although hot holes have a strong transport capacity, their short duration is averaged over the time domain by subsequent long periods of slow hopping transport, resulting in a gap in the measured values.

## Unveiling the delocalized hot carriers in $Co_3O_4$ using ultrafast THz photoconductivity

To further elucidate the electrical transport properties of charge carriers, we employed THz spectroscopy to directly monitor the photoconductivity of $Co_3O_4$ film (Fig. 3a). For optical-pump-THz-probe (OPTP) measurements, the transient photoconductivity was detected

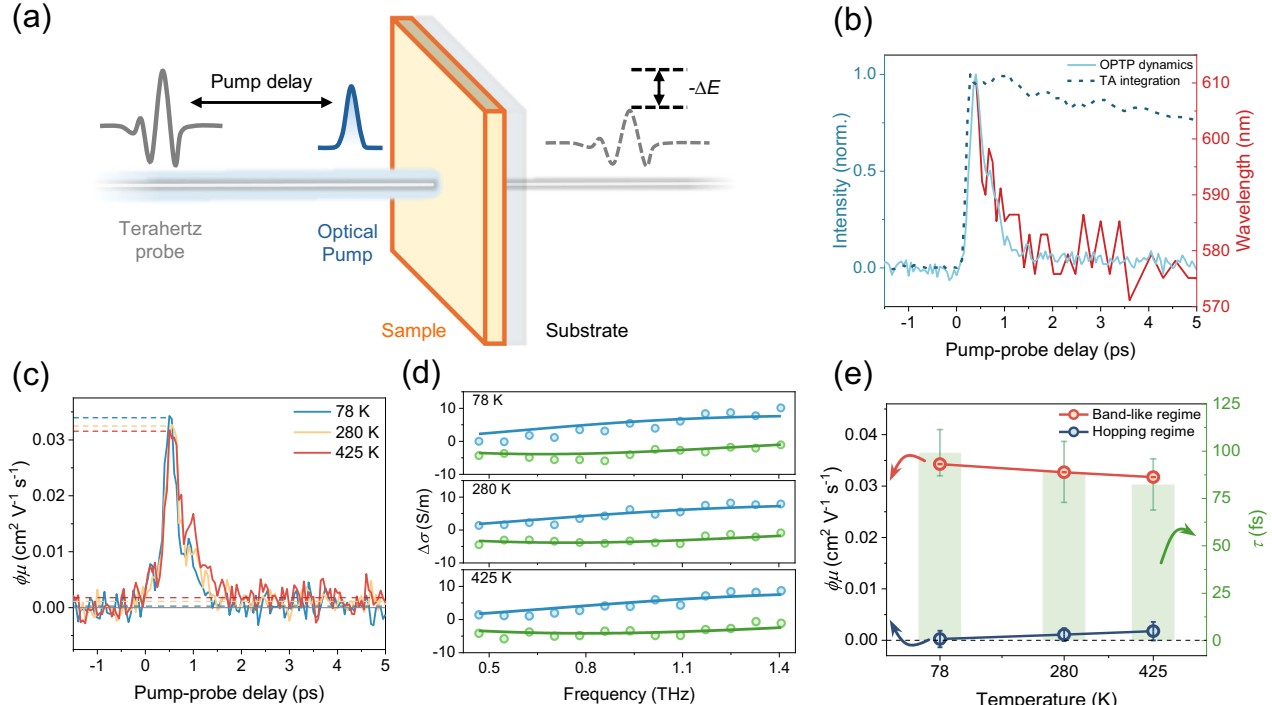

**Fig. 3 | THz spectroscopy measurements of Co₃O₄. a** Schematic illustration of THz measurement. **b** Normalized OPTP dynamics at 3.1 eV pump (light blue solid line), normalized integrated TA signal (dark blue dashed line) obtained by summing the PIA signal within 550–650 nm at each delay time, representing the total hole population including both free and localized species, and blue shift of the peak of ~600 nm PIA signal. (red solid line) **c** Temperature dependence of the effective carrier mobilities, following excitation at 3.1 eV. The dashed line is used as a guide and represents the effective carrier mobility values for the two transport regimes (peak value for hot carrier transport regime and average value after 2 ps for polaron hopping transport). **d** The complex frequency-resolved photoconductivity was measured at a pump-probe delay time of 0.5 ps. The blue and green dots represent the real and imaginary parts of the measured complex photoconductivity, respectively. The corresponding solid line represents a Drude-Smith description. **e** Effective carrier mobility in the band-like regime (red) and hopping regime (blue), and the inferred charge scattering time $\tau$ (green) of the band-like transport regime from the Drude-Smith model. Error bars represent standard deviations from 300 scans (hot-carrier transport), standard deviations of 2–5 ps averages (polaron hopping), and Drude-Smith fitting uncertainties for charge scattering time (shown in green).

by fixing the sampling beam to the peak of the terahertz pulse. The relative attenuation of the terahertz electric field ($-\triangle E/E$) induced by the pump is proportional to the transient photoconductivity $\sigma$ (=$n \cdot e \cdot \mu$, with $n$ is the density of photogenerated charge carriers, $\mu$ is the charge mobility, and $e$ is the elementary charge)[47,48]. OPTP results revealed a rapid increase in photoconductivity within a few hundred femtoseconds, followed by a swift decline to a long-lived, small, but non-zero photoconductivity plateau within the next ps (Fig. 3b). The photoconductivity dynamics agree remarkably well with the blue shift in the TA spectrum. Furthermore, the slow decay of the integrated dynamics from 550 nm to 650 nm in the TA spectra excludes the carrier recombination as the primary cause of the decline in photoconductivity. These results corroborate the notion that the reduction in photoconductivity is due to the formation of localized polaron states with low mobility (Supplementary Note 10), and our assignment of the blue shift in the TA spectra as the signature of polaron formation. More detailed, comparative analysis based on ultrafast optical nanoscopy and THz spectroscopy is discussed in Supplementary Note 11.

We further performed temperature-dependent OPTP measurements spanning from 425 K to 78 K to confirm the carrier transport and scattering mechanism in Co₃O₄ (Fig. 3c). Here, we present the time-dependent product of $\phi\mu$ (where $\phi$ is the free carrier generation quantum yield and $\mu$ is the mobility; see details for the measurements in Supplementary Note 12 and 13). By lowering $T$, we observe an intriguing difference between the $T$-dependent photoconductivity peak and plateau: the peak conductivity goes up with lowering $T$, in line with the band transport behavior of delocalized hot carriers. In

contrast, the photoconductive plateau decreases with lowering $T$, as expected for hopping transport of polaronic states. In the band-like transport regime, as temperature decreases, lattice vibrations and thus phonon-induced scattering are reduced, which favors charge transport. In contrast, in the hopping transport regime, carrier (i.e., polarons) conduction requires thermal excitations, leading to reduced hopping probabilities at low temperatures[49,50].

In order to better understand the carrier scattering mechanism of the highly mobile hot carriers in Co₃O₄, frequency-resolved complex photoconductivity measurements of Co₃O₄ thin films were carried out in the same temperature range using Terahertz time-domain spectroscopy at a pump-probe delay of 0.5 ps. In all cases, the frequency-resolved complex photoconductivity ($\sigma(\omega)$) is well described by the Drude-Smith model (Fig. 3d), as in the following equation[48,51]:

$$\sigma(\omega) = \frac{\omega_p^2 \varepsilon_0 \tau}{1 - i\omega\tau}\left(1 + \frac{c}{1 - i\omega\tau}\right) \tag{1}$$

where $\omega_p$, $\varepsilon_0$, $\omega$, $\tau$, and $c$ represent the plasma frequency, vacuum permittivity, angular frequency, carrier scattering time, and additional parameters for describing the probability of scattering events, respectively. Figure 3e shows the effective mobility of the two transport regimes corresponding to Fig. 3c, and the carrier scattering time of the band-like regime extracted from the Drude-Smith fit as a function of temperature. The inferred carrier scattering time ($\tau$) displays a very similar $T$-dependence as the peak value of $\phi\mu$: both $\tau$ and $\phi\mu$ increase with decreasing temperature, supporting the delocalized charge nature of these short-lived hot carriers.

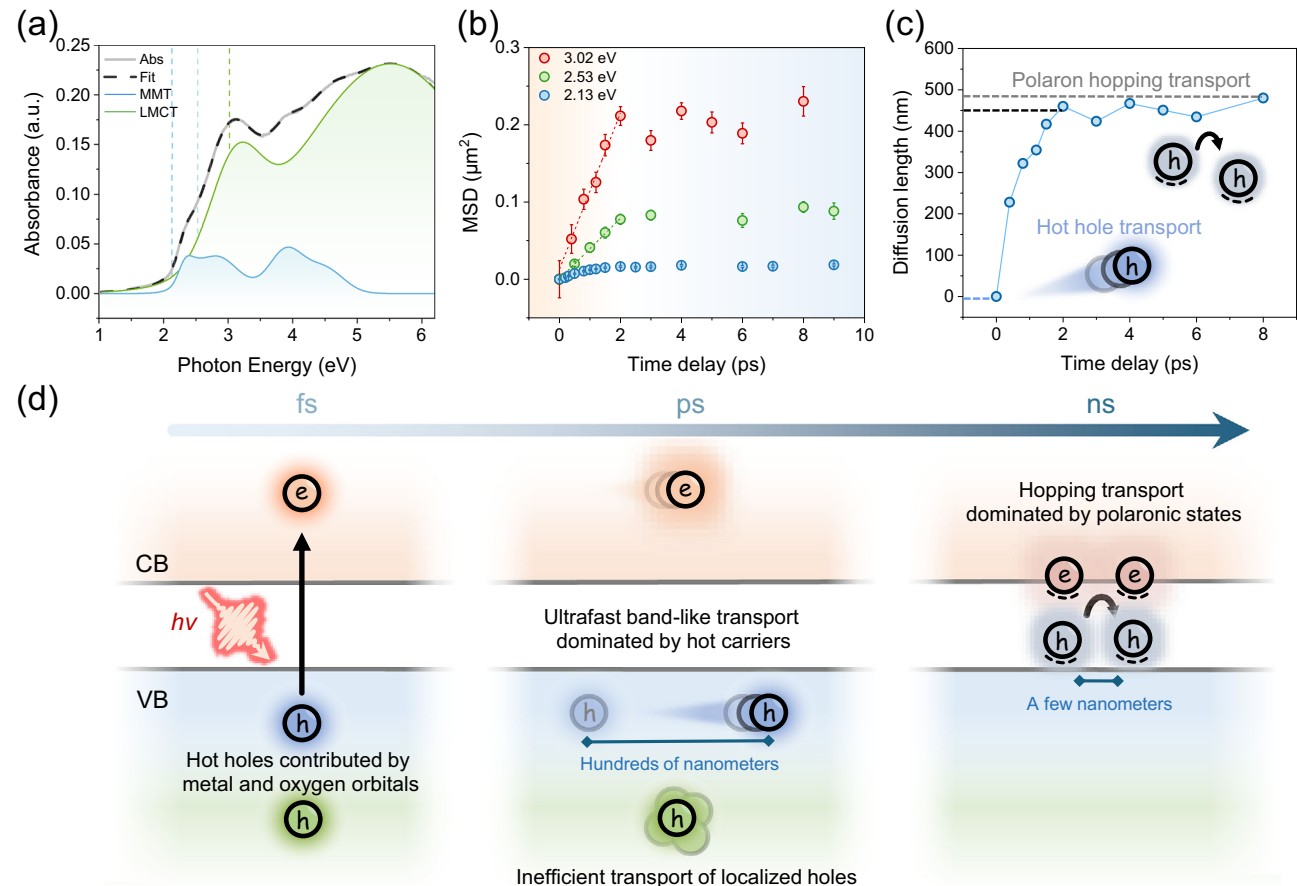

**Fig. 4 | Spatial dynamics of photogenerated holes in α-Fe₂O₃. a** Steady-state absorption spectrum of the α-Fe₂O₃ film. The spectrum was fitted (black dash line) to the sum of seven Gaussian bands. The absorption bands assigned to LMCT transitions are shown in green, and those assigned to MMT are shown in blue. The vertical dashed lines indicate the pump photon energies used in the ultrafast optical nanoscopy measurements. **b** MSD evolution of carrier population distribution for α-Fe₂O₃ at near-bandgap (2.13 eV) and above-bandgap (2.53 eV, 3.02 eV) excitations with delay time. Error bars represent 2D Gaussian fitting errors. **c** The relationship between hole diffusion length and delay time of α-Fe₂O₃ at 3.02 eV pump photon energy. **d** Schematic diagram of carrier transport in TMOs.

## Investigation of hot carrier diffusion in α-Fe₂O₃

Given the shared characteristics of d-orbital electronic structures in TMOs, we extended our investigation to another open d-shell TMO, α-Fe₂O₃, performing similar ultrafast optical nanoscopy measurements to demonstrate the general applicability of ultrafast hot hole transport in TMOs. The steady-state absorption spectrum of α-Fe₂O₃ is shown in Fig. 4a, and absorption bands are assigned according to previous literature[52] (More details in Supplementary Note 14). The absorption spectrum indicates a bandgap of about 2.1 eV for the α-Fe₂O₃ film. The TA spectrum of the α-Fe₂O₃ thin film can be found in Supplementary Note 15. For ultrafast optical nanoscopy experiments, pump photon energies for near-bandgap (2.13 eV) and above-bandgap (2.53 eV, 3.02 eV) excitations were selected, using low pump fluence to avoid higher-order effects that may impact the data analysis (Supplementary Note 16). A probe wavelength of 675 nm was selected to monitor the dynamics of band-edge holes based on the characteristic hole signal wavelength obtained from the bias-dependent TA study of Barroso et al.[52,53]. Similar to Co₃O₄, the hot holes of α-Fe₂O₃ also exhibit remarkably efficient transport (Fig. 4b and Supplementary Note 17), lasting for about 2 ps, after which the hole combines with the small electron polaron to form a self-trapped exciton by coulomb attraction based on previous TA study[54]. In addition, we observed highly mobile hot holes in both α-Fe₂O₃ as a charge-transfer semiconductor and Co₃O₄ as a Mott-Hubbard semiconductor, underscoring the universality of ultrafast hot hole transport in open d-shell TMOs.

While for both studied TMO materials, hot holes have been shown to possess higher diffusivity than their cold counterparts, we further demonstrate that the hot carrier diffusion lengths in the two TMOs exhibit distinctly different dependence on pumping hυ. In the case of α-Fe₂O₃, compared to MMT-dominated excitation (by 2.53 eV), LMCT-dominated excitation (3.02 eV) leads to a higher diffusion constant of up to $510 \pm 33$ cm² s⁻¹ and a diffusion length of approximately 500 nm. This result indicates an apparent increase of hot carrier diffusion length by increasing hυ, being consistent with the photocurrent response of α-Fe₂O₃ in the literature[18,23]. Importantly, we note that near the absorption edge of α-Fe₂O₃, the optical response is primarily governed by localized ligand field (LF) transitions within Fe³⁺ ions. These transitions are intrametallic and non-charge-transfer in nature, and their strongly localized character may account for the lower diffusion coefficients observed under lower-energy excitation (Supplementary Note 14). Similar to Co₃O₄, the hot holes in α-Fe₂O₃ also account for more than 90% of the total diffusion length for mobile carriers in merely 2 ps, as shown in Fig. 4c. Complementing transient transport measurements, photocurrent measurements of α-Fe₂O₃ photoanodes show consistent results: Photocurrent measurement on α-Fe₂O₃ films by Kay and colleagues using a solar simulator demonstrate that, with excitation above the bandgap, carriers exhibit diffusion over several hundred nanometers, and longer diffusion lengths are observed at higher excitation photon energies[17]. It is important to emphasize that such ultrafast, long-range diffusion occurs only for photocarriers excited via delocalized optical transitions (Fig. 4d). As such, our findings, together with photocurrent measurements, validate that long-distance hot hole transport is an intrinsic property of α-Fe₂O₃ and highlight its significant potential for practical applications.

Finally, we note that while our transport measurements align well with the photocurrent performance of α-Fe$_2$O$_3$, it challenges the widely accepted assumption, based on the Gartner model, that the diffusion length in α-Fe$_2$O$_3$ photoanodes is predicted to be only 2–4 nm by simulation[15]. The current theoretical model has not yet taken the contribution of non-equilibrium carriers into account (which by no means will be easy), leading to a potential underestimation of the total carrier diffusion length. Additionally, in optoelectronic transport measurements lacking high temporal resolution, the short-lived process of hot hole transport is often obscured by longer, slower polaronic hopping transport. Previous THz spectroscopy study of α-Fe$_2$O$_3$ has shown a rapid decrease in photoconductivity at 2 ps after excitation, corroborating our findings[55]. In practical scenarios where most incident photons have energies exceeding the bandgap, the generation of hot carriers is inevitable. To further evaluate their impact on the photocurrent response, we performed numerical simulations of carrier collection efficiency in a 500 nm-thick hematite film using the transport parameters from this work (Supplementary Note 18). The results show that under 2.53 eV excitation, approximately 2.15% of carriers can be collected, while under 3.02 eV excitation, the collection efficiency increases to 9.46%, in good agreement with the wavelength-dependent photocurrent trend observed experimentally. More importantly, in both cases, hot carriers account for over 70% of the total collected carriers. Given their substantial contribution to the overall diffusion length, neglecting their impact could lead to inaccurate assessments of carrier transport properties, potentially hindering the optimal design of PEC systems[56].

## Discussion

Beyond unveiling hot holes with high diffusivity in TMOs, our study also highlights that the nature of optical transitions, rather than just the excess energy of hot carriers, dominates their diffusion. However, one puzzle remains: how to rationalize the role of chemical composition (α-Fe$_2$O$_3$ vs Co$_3$O$_4$) in impacting optical transitions and further hot carrier diffusivity. For instance, in contrast to α-Fe$_2$O$_3$, the primary free carriers in Co$_3$O$_4$ originate from MMT, rather than LMCT. While the precise mechanism remains unclear, we propose that this discrepancy may originate from the difference in the localization pathways following distinct optical transitions. This effect, together with the mobility differences between the initial and final electronic states involved, jointly determine both the time window and the rate at which hot carriers can retain high mobility prior to self-trapping. Specifically, the balance between the on-site Hubbard repulsion ($U$) and the superexchange interaction ($J$) plays a critical role in dictating carrier localization[57]. Different excitation pathways perturb this $U$–$J$ balance in opposite directions, either enhancing or suppressing polaron formation. Given the significant role that Hubbard repulsion plays in polaron formation, the differences in hot carrier transport observed in these two materials may reflect the fundamental differences in the electronic structure between charge transfer semiconductors (such as α-Fe$_2$O$_3$) and Mott-Hubbard semiconductors (such as Co$_3$O$_4$).

Beyond polaron formation, another plausible explanation involves the interaction between optical excitations and mid-gap localized ligand-field (LF) states, as recently proposed by Sachs et al.[58] In open d-shell TMOs such as α-Fe$_2$O$_3$ and Co$_3$O$_4$, intrametallic LF states introduce efficient sub-picosecond nonradiative decay pathways. In Co$_3$O$_4$, LMCT transitions are energetically close to these LF states and are spin-allowed, which facilitates ultrafast carrier relaxation and strong localization, thereby suppressing diffusion. In contrast, MMT transitions may partially bypass these LF-mediated decay channels, enabling more efficient carrier transport. In α-Fe$_2$O$_3$, by comparison, relaxation from LMCT to LF states is spin-forbidden due to optical selection rules. As a result, carriers generated via LMCT must undergo spin-flip processes before being trapped into LF states,

delaying localization and thereby extending the lifetime of delocalized transport.

Given the diversity of electronic structures across transition metal oxides, including variations in d-electron occupancy, it is essential to assess whether the ultrafast transport phenomena observed here extend beyond open-shell systems. For that, we further studied the carrier diffusion effects in Cu$_2$O, a d$^{10}$-configured TMO with fully occupied metal d orbitals. We observed comparable picosecond-scale diffusion behavior (Supplementary Note 19), indicating that ultrafast hot-carrier transport can also occur in closed d-shell oxides. This trend extends to more complex systems as well. For example, BiVO$_4$, a representative ternary TMO with an empty d-shell, has recently been reported to exhibit spatially selective distributions of electrons and holes within the first few picoseconds after photoexcitation, supporting the presence of non-equilibrium hot-carrier transport in the early-time regime[59]. Taken together, these findings suggest that ultrafast hot-carrier diffusion is likely a general phenomenon across a broad range of TMOs.

In addition to revealing the ultrafast, long-range diffusion of transient mobile carrier states, our study provides further non-conventional pathways for improving device PEC performance by exploiting hot carrier effects. For instance, when combined with a photon up-conversion scheme, we can activate high-energy delocalized transitions that are typically inaccessible under below-bandgap low-energy Infrared excitations. Furthermore, the highly diffusive nature of hot carriers partially relaxes the hot carrier extraction time window across hybrid interfaces, which holds great promise for increasing the attainable open-circuit voltage in optoelectronic systems or enabling chemical reactions that are otherwise difficult to realize under thermal equilibrium conditions. Overall, this spatio-temporal dynamic coupling ensures both efficient spatial delivery of charge carriers to reactive sites and their prolonged retention at these locations, providing a practical route toward enhancing charge utilization in photoelectrochemical energy conversion.

In summary, our work offers a comprehensive description of carrier transport in two representative TMOs (Co$_3$O$_4$ and α-Fe$_2$O$_3$), highlighting the significant transport capabilities of hot holes generated by delocalized optical transitions. Selective excitation of different optical transitions unveils the importance of optical transitions, rather than merely energetics, in dictating hot hole transport. While each TMO has shown uniqueness in electronic and optical properties, we demonstrate that ultrafast band-like transport of hot carriers accounts for the major portion of the total diffusion length in both Co$_3$O$_4$ and α-Fe$_2$O$_3$. Therefore, the pivotal contribution of hot carriers must be considered when evaluating the diffusion length of photogenerated carriers in TMOs. Optical transition types, as the key to unlocking efficient hot carrier transport, combined with the intrinsic long-lived polarons in TMOs, offer a fascinating platform for designing advanced TMO-based optoelectronic devices that leverage photoexcited states for efficient energy conversion.

## Methods

### Sample preparation

Co$_3$O$_4$ thin film sample was prepared on a quartz substrate by pulsed laser deposition. The laser repetition frequency was set to 5 Hz, the working temperature to 500 °C, the working pressure to 15 Pa, and the deposition time to 30 min in an oxygen atmosphere. The sample for THz measurements was prepared by magnetron sputtering. Initially, Co$_3$O$_4$ was added to a 5% aqueous solution of PVA-205 and stirred. The mixture was then transferred to a mold and cold pressed at 30 MPa for 30 minutes, followed by annealing at 800 °C for 10 h in the air, with a heating rate of 5 °C per minute. During the sputtering process, the power was set to 100 W and the sputtering duration was 2 h, followed by annealing in air at 600 °C for 2 h. α-Fe$_2$O$_3$ thin film was grown on a

sapphire substrate. 99.99% pure $Fe_2O_3$ powder was sintered at 1250 °C in air for 12 h and used as the target material. A Krypton fluoride excimer laser ($\lambda = 248$ nm) with a repetition rate of 5 Hz was used for ablating the target. The energy density of the laser irradiation was 1.2 J cm$^{-2}$. The ablated material was deposited on a sapphire substrate, which was heated to 550 °C during the growth process in an oxygen background pressure of 100 mTorr. The film was then cooled to room temperature under the same $O_2$ background pressure.

## Ultrafast TA spectroscopy

The femtosecond transient absorption spectrometer consists of two main components: a regenerative-amplified Ti: sapphire laser system (Coherent) and the Helios pump-probe system (Ultrafast Systems). The regenerative-amplified Ti: sapphire laser system (Legend Elite-1K-HE), operated at a center wavelength of 800 nm, had a pulse duration of 25 fs, a pulse energy of 4 mJ, and a repetition rate of 1 kHz. The 800 nm fundamental output of the amplifier was split into two separate beam pulses. The majority of the fundamental beam passed through the optical parametric amplifiers (TOPAS-C). The resulting output light serves as the pump light and is modulated by a mechanical chopper operating at a frequency of 500 Hz. Another portion of the fundamental beam was focused onto a sapphire or YAG crystal after passing through a motorized optical delay line to generate a white light continuum as the probe light. The optical path difference, controlled by the motorized optical delay line, between the pump light and the probe light was utilized to monitor transient states at various pump-probe delay times. A reference beam was extracted from the white light continuum to correct pulse-to-pulse fluctuations within the white light continuum. The pump and probe beams were spatially and temporally overlapped on the sample.

## Ultrafast optical nanoscopy

For the ultrafast optical nanoscopy measurements, a schematic layout of the optical setup is provided in Supplementary Note 5. Briefly, ultrafast optical nanoscopy measurements use a high repetition rate amplifier (PH1-20, Light Conversion, 800 kHz, 1030 nm) as the light source. Two independent outputs are generated by optical parametric amplifiers (OPA, TOPAS-Twins, Light Conversion). One output served as the pump source, while the other acted as the probe source. Both the pump and probe beams underwent spatial filtering. Modulation of the pump beam at 100 kHz was achieved using an acousto-optic modulator (Gooch and Housego, AOMO 3080–125). To introduce a delay between the pump and probe, a mechanical linear motor stage (M-IMS600LM-S, Newport) was employed. An objective (MRH08430, Nikon, 40×, NA = 0.6) was used to focus both the pump and probe beams onto the sample, with the transmitted probe beam collimated by an aspherical lens and subsequently detected by an avalanche photodiode (APD; Thorlabs APD430A/M). A phase-locked amplifier (HF2LI, Zurich Instruments) is used to detect changes in the probe signal caused by the pump. Spatial scanning of the probe beam relative to the pump beam was accomplished with a two-axis Galvo mirror (GVS012/M, Thorlabs) to obtain images depicting the distribution of carriers.

## Terahertz spectroscopy

For the terahertz measurement, a commercial, regenerative amplified mode-locked Ti: Sapphire laser system provides femtosecond pulses with a duration of about 50 fs, a central wavelength of 800 nm, and operates at a repetition rate of 1 kHz. The fundamental 800 nm laser pulses are divided into three branches for optical excitation, terahertz generation, and electro-optical sampling. The 400 nm pump pulses used for optical excitation are generated by second harmonic generation using a beta-barium borate (BBO) crystal, and other wavelengths are obtained by commercial optical parametric amplifiers from Light Conversion. For THz generation, ~10% of the

800 nm laser pulses are used to produce single-cycle THz pulses with a duration of ~1 ps and a bandwidth of 1 THz by optical rectification via a 1 mm thick ZnTe (110) crystal. The generated terahertz pulses are focused on the sample using a pair of 90° off-axis parabolic mirrors and are detected through time-resolved electro-optic sampling. Room temperature terahertz measurements were conducted in a dry $N_2$ atmosphere, while low-temperature terahertz measurements were carried out in vacuum ($p < 10^{-4}$ mbar) by mounting the sample inside a cryostat.

## Data availability

All the relevant data that support the findings of this work are available from the corresponding author upon request. Source data are provided with this paper.

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

## Acknowledgements

This work was supported by the Scientific Research Innovation Capability Support Project for Young Faculty (ZYGXQNJSKYCXNLZCXM-D3), Basic Science Center Program of the National Natural Science Foundation of China (Grant No.52488301), Beijing Natural Science Foundation (JQ23009), National Key R&D Program of China 2024YFB4607900, and

the BIT Research and Innovation Promoting Project (Grant No. 2024YCXZO18). K.L. acknowledges the Max Planck Graduate Center (MPGC) for funding support during the research stay in Germany. G.W. acknowledges the fellowship support from the China Scholarship Council (CSC). K.L. also appreciates Lei Gao and Lucia Di Virgilio at Max Planck Institute for Polymer Research for their assistance with THz measurements, and Longren Li and Jinshui Cheng for their valuable experimental assistance. We acknowledge Yujin Tong at University Duisburg-Essen for fruitful discussions.

## Author contributions

K.L., Y.W., H.I.W., and T.Z. conceived the project. Y.Z., X.W., and S.L. contributed to the sample synthesis. K.L. and G.W. performed the THz measurements and analyzed the results under the supervision of H.I.W. K.L., Y.W., and G.G. conducted the ultrafast optical nanoscopy measurements and analyzed the data under the supervision of T.Z. and L.J. K.L., Y.W., M.B., H.I.W., and T.Z. wrote the manuscript with input from all authors. All authors discussed the results and commented on the manuscript.

## Competing interests

The authors declare no competing interests.
