## [Transparent Peer review file · Nature Communications]

Unlocking Ultrafast Hot Hole Transport in Transition Metal Oxides Governed by the Nature of Optical Transitions

Corresponding Author: Professor Tong Zhu

Version 0:

Reviewer comments:

Reviewer #1

(Remarks to the Author)

The article by Li et al investigates the role of optical transitions on carrier dynamics in two transition metal-oxides, hematite and Co_3O_4 using a combination of ultrafast spectroscopic techniques. The main result is the identification of two distinct hole transport regimes, one with ultrahigh diffusivity and one with significantly lower diffusivity dominated by small-polaron dominated transport.

I read this article with great interest. In recent years, the importance of optical transitions in determining carrier dynamics in open d-shell transition metal oxides (TMOs) has become increasingly evident. This manuscript is timely, provides significant mechanistic insight into this phenomenon, and merits publication in Nature Communications provided the authors can address the following points:

1) The manuscript categorizes only two types of electronic transitions, ligand-to-metal charge transfer (LMCT) and metal-to-metal charge transfer (MMCT), both of which are charge-transfer in nature. The authors suggest that MMCT bands, in addition to LMCT bands, contribute significantly to the absorption spectrum of hematite. However, numerous studies, including those cited by the authors, attribute these features to localized ligand field (LF) transitions, which are intrametallic (i.e., involve the same metal center) and non-charge-transfer in nature. LF transitions are much more localized than MMCT and LMCT transitions and may more accurately account for the lower diffusivity observed. This alternative interpretation may be more consistent with the wavelength-dependent behavior between Co_3O_4 , where MMCT transitions at longer wavelengths generate highly mobile carriers, and hematite, where more localized LF transitions dominate the response at long wavelengths.

2) The authors should show 2D maps (similar to those shown in Figure 2) from the nanoscopy measurements for both materials, and under different pump wavelengths.

3) Given the spectral overlap between the various absorption bands, excitation at a fixed pump wavelength likely results in the generation of multiple types of excited carriers (e.g. via MMCT + LMCT or LMCT + LF). The resulting signals are therefore not purely attributable to a single type of excitation. The authors should clarify how they account for this overlap and whether deconvolution was performed to isolate the contributions from each transition type.

4) The question of whether highly mobile, band-like carriers are responsible for the primary photocurrents in hematite-based photoelectrochemical (PEC) devices remains unresolved. For instance, the authors cite work by Kay et al., which reports a quantum efficiency of only ~2% for carriers capable of long-range transport. Can the authors comment on the yield of these "hot" carriers in their experiments as a function of pump wavelength? Such an analysis would be valuable for assessing the potential device relevance of the observed fast transport regime.

Reviewer #2

(Remarks to the Author)

This work presents a description of carrier transport in two different TMOs (Co_3O_4 and $\alpha\text{-Fe}_2\text{O}_3$) and decoupled the effect of hot hole transport and polaron hopping dynamics in these two materials. Femtosecond temporal and spatial resolution

probing was utilized. The authors showed that these two materials have different behavior depending on the excitation energy and transition type, showing material specific carrier transport regimes. In my opinion, this work provides insightful interpretation of carrier and diffusion characteristics in TMOs. I recommend minor revisions before publication.

1. The author mentioned that in case of α -Fe₂O₃, the diffusion constant is higher for LMCT-dominated excitation (3.02eV) than MMCT-dominated excitation (2.53eV), which is an opposite trend to Co₃O₄ case. In case of Co₃O₄, the diffusion constant is lower for LMCT-dominated excitation (2.68eV) than MMCT-dominated excitation (1.55eV). Can the author describe more explicitly why different TMOs have different diffusion constant behavior for the carrier generation mechanisms?
2. In Figures 1a and 4a, the MMCT and LMCT appear in different colors, which is confusing.
3. In Figure 3b, can author describe more detail how the TA integration was obtained?
4. The authors describe their ultrafast optical measurement as "Nanoscopy". The optical setup in Figure S8 simply utilizing far-field optics whose resolution is set by diffraction limits. The author needs to justify the use of the term, or needs to use an alternative.
5. How about other TMOs? Can the author provide an overall insight for the carrier dynamics and behavior of various other TMOs?
6. What is the TMO film thickness? Did the author account for the effect of film thickness (in terms of optical, carrier scattering effects)?

Reviewer #3

(Remarks to the Author)

This manuscript presents a comprehensive study on ultrafast hot hole transport in transition metal oxides (TMOs), specifically Co₃O₄ and α -Fe₂O₃, using ultrafast optical microscopy and terahertz spectroscopy. However, several key concerns about the interpretation of spectral features and dynamics must be addressed before publication.

1. The assignment of the 600 nm PIA to "photoinduced hole absorption from O 2p orbitals to the valence band maximum" (Lines 163–164) lacks direct experimental evidence. On the other hand, this feature has been assigned to the onsite d-d transition of the Co²⁺ ions in earlier works.
2. In Fig. 1c, the kinetic trace probed at 600 nm shows distinct difference compared with those probed at 435nm and 910 nm. However, the difference kinetic trend cannot justify that the 600 nm PIA is associated with hole polarons. As shown in Fig. 1d, the d-d transition feature overlaps with the BGR induced positive feature, and this spectral overlap may account for a different kinetic trend at 600 nm.
3. There may be an alternative interpretation for the apparent blue shift at 600 nm. As a consequence of the decay of BGR feature (PIA beyond 600 nm), the d-d transition centered at ~575 nm remains, giving rise to an apparent blue shift. Thus, there should be two different dynamics in this system: one is the sub-ps carrier localization due to the self-trapping. The other one is the recombination of the self-trapped excitons.
4. Given the fact of the "hot carriers" with sub-ps lifetime, the end of the THz pulse with a ps time cycle will experience a different response than the beginning of the pulse, which will complicate the analysis. Kindt and Schmuttenmaer had carefully evaluated this issue in J. Chem. Phys. 1999, 110 (17), 8589-8596. The author should provide a detail procedure showing that the mobility of transient hot carriers can be extracted by the THz spectroscopy.
5. The sub-ps carrier localization shown by the THz kinetics is attributed to the hot carrier relaxation. However, the self-trapping of the photocarriers can give rise to the same decay dynamics. Can the self-trapping be excluded to account for the fast THz decay?
6. This manuscript claims that the hot carrier diffusion coefficient depends on the pump photon energy (as shown in Fig. 2d and Fig. 4b). Why the MSD for Fe₂O₃ and Co₃O₄ show different dependence on pump photon energy?

Version 1:

Reviewer comments:

Reviewer #1

(Remarks to the Author)

The authors have addressed my comments. I recommend publication

Reviewer #2

(Remarks to the Author)

The author has addressed the reviewers' comments appropriately. I recommend acceptance.

Reviewer #3

(Remarks to the Author)

The authors have addressed all the raised issues. I recommend acceptance.

REVIEWER COMMENTS

Reviewer #1 (Remarks to the Author):

The article by Li et al investigates the role of optical transitions on carrier dynamics in two transition metal-oxides, hematite and Co₃O₄ using a combination of ultrafast spectroscopic techniques. The main result is the identification of two distinct hole transport regimes, one with ultrahigh diffusivity and one with significantly lower diffusivity dominated by small-polaron dominated transport.

I read this article with great interest. In recent years, the importance of optical transitions in determining carrier dynamics in open d-shell transition metal oxides (TMOs) has become increasingly evident. This manuscript is timely, provides significant mechanistic insight into this phenomenon, and merits publication in Nature Communications provided the authors can address the following points:

Response: We appreciate reviewer 1's positive comments and careful review on our work. In the revised manuscript, we clarified the classification of optical transitions (LMCT, MMCT, LF), provided numerical modeling of wavelength-dependent carrier extraction efficiency, further discussed spectral assignments and technical details, and discussed broader hot carrier transport implications in TMOs. We believe that our revisions in response to the reviewer's suggestions have substantially improved the manuscript.

1) The manuscript categorizes only two types of electronic transitions, ligand-to-metal charge transfer (LMCT) and metal-to-metal charge transfer (MMCT, both of which are charge-transfer in nature. The authors suggest that MMCT bands, in addition to LMCT bands, contribute significantly to the absorption spectrum of hematite. However, numerous studies, including those cited by the authors, attribute these features to localized ligand field (LF) transitions, which are intrametallic (i.e., involve the same metal center) and non-charge-transfer in nature. LF transitions are much more localized than MMCT and LMCT transitions and may more accurately account for the lower diffusivity observed. This alternative interpretation may be

more consistent with the wavelength-dependent behavior between Co_3O_4 , where MMCT transitions at longer wavelengths generate highly mobile carriers, and hematite, where more localized LF transitions dominate the response at long wavelengths.

Response: We sincerely thank the reviewer for raising this important question concerning the ligand field (LF) transitions. We fully agree that local LF transitions can indeed play a crucial role in shaping carrier transport in both Co_3O_4 and hematite. For instance, recent study by Sachs et al. (*Nat. Chem.* 2025, <https://doi.org/10.1038/s41557-025-01868-y>) have revealed that LF states can introduce sub-picosecond carrier localization channels in TMOs, a mechanism that we have discussed in detail in the *Discussion* section in the main text.

While the effect of LF transitions is clearly reported, transitions occur mostly at the band edge. Taking hematite as an example, the absorption edge is generally attributed to the LF transitions within Fe^{3+} ions. At higher photon energies, pair LF transitions have been reported across a rather broad range (~ 3 eV to 5.6 eV), significantly overlapping with LMCT transitions and thereby complicating precise spectral assignments (*Phys. Rev. B* 2005, 71(12): 125411; *J. Phys. Chem. C* 2011, 115(42): 20795-20805; *Phys. Chem. Chem. Phys.* 2010, 12, 14045-14056). Additionally, Su et al. suggest that although pair LF transitions do not directly generate free electron-hole pairs, they may form Fe^{2+} - Fe^{4+} charge pairs via superexchange or hopping mechanisms, exhibiting the characteristics of charge transfer transitions (*J. Am. Chem. Soc.* 2017, 139, 4916–4922). Furthermore, Chernyshova et al. observed that when high-energy LF transitions are mixed with overlapping CT absorption, they may exhibit partial delocalization, which further obscures their localized nature and precise spectral assignment (*Phys. Chem. Chem. Phys.* 2010, 12, 14045-14056).

Overall, LF transitions could play an important role in absorption (and subsequently charge transport), but mostly near the band edge, where absorption is weak. Their relative contribution diminishes at higher photon energies due to stronger selection rule constraints, with LMCT transitions instead dominating the absorption cross-section. Therefore, we did not provide further discussion on LF transitions above the band gap in the original manuscript. However, in this revision, in response to the comment from the reviewer, we have included a discussion on the potential impact of near-band-edge LF transitions on the wavelength-dependent carrier

transport behavior in hematite, together with the underlying physical rationale.

A similar situation holds for Co_3O_4 . The near-band-edge absorption feature (~ 0.83 eV) is commonly attributed to LF transitions within Co^{2+} ions. However, the presence and position of LF transitions for Co^{3+} ions in the visible range remains much debated. For example, Lima et al. argued that the visible absorption of Co_3O_4 mainly arises from LF transitions of tetrahedral Co^{2+} and octahedral Co^{3+} rather than interionic charge transfer (*J. Phys. Chem. Solids* 2014, 75(1): 148-152). Other studies suggest that the ~ 1.9 eV absorption peak may involve overlap between MMCT and LF transitions (*J. Mater. Chem. C* 2013, 1, 4628-4633). Conversely, Kim et al. suggest that the strong visible absorption of Co_3O_4 is dominated by MMCT and LMCT, with LF transitions playing a negligible role (*Solid State Commun.* 2003, 127(1): 25-28). Given these different interpretations, we believe that it remains difficult to definitively assign high-energy LF transitions at this stage. However, this does not affect our main conclusions about carrier transport mechanisms. If localized high-energy LF transitions do exist, they would typically have excitation energies higher than MMCT but lower than LMCT, which is consistent with our experimental observation that higher-energy excitation tends to reduce carrier diffusion due to stronger localization.

Action: To improve clarity and accuracy, we use the term “metal-to-metal transitions (MMT)” to replace “metal-to-metal charge transfer (MMCT)” throughout the manuscript and Supplementary Information. The sentence in the introduction “metal-to-metal charge transfer (MMCT)” was changed to “metal-to-metal transitions (MMT) originating within or between metal d orbitals”. Following the wavelength-dependent transport measurements of hematite, we added: “Importantly, we note that near the absorption edge of $\alpha\text{-Fe}_2\text{O}_3$, the optical response is primarily governed by localized ligand field (LF) transitions within Fe^{3+} ions. These transitions are intrametallic and non-charge-transfer in nature, and their strongly localized character may account for the lower diffusion coefficients observed under lower-energy excitation (Supplementary Note 14).” We also included a discussion in the Supplementary Note 14 to further clarify the potential contribution of the LF transition in steady state absorption.

2) The authors should show 2D maps (similar to those shown in Figure 2) from the nanoscopy

measurements for both materials, and under different pump wavelengths.

Response and action: We thank the reviewer for this valuable suggestion. We have added the 2D maps from nanoscopy measurements of Co_3O_4 and hematite in Supplementary Note 8 and Supplementary Note 17, respectively

Figure R1.1: 2D spatial maps of ultrafast optical nanoscopy images of Co_3O_4 at representative time delays under pump photon energies of (a) 2.58 eV, (b) 2.30 eV, (c) 1.94 eV, and (d) 1.55 eV.

Figure R1.2: 2D spatial maps of ultrafast optical nanoscopy images of hematite at representative time delays under pump photon energies of (a) 3.02 eV, (b) 2.53 eV, and (c) 2.13 eV.

3) Given the spectral overlap between the various absorption bands, excitation at a fixed pump wavelength likely results in the generation of multiple types of excited carriers (e.g. via MMCT + LMCT or LMCT + LF). The resulting signals are therefore not purely attributable to a single type of excitation. The authors should clarify how they account for this overlap and whether deconvolution was performed to isolate the contributions from each transition type.

Response: We appreciate the reviewer's comment. We agree that due to the significant spectral overlap among different optical transitions, excitation at a fixed pump wavelength can indeed generate multiple types of excited carriers, resulting in transient signals that represent a combined response from different excitation pathways. In this case, both metal holes and oxygen holes may be simultaneously involved, contributing jointly to the observed transient

characteristics.

In our nanoscopy measurements, the mean square displacement (MSD) reflects the spatial expansion of the hole population, allowing us to extract an effective diffusion coefficient. It should be emphasized that this diffusion coefficient is, by nature, a weighted average of the diffusion rates of multiple excited hole species. Its magnitude depends on the relative proportions of localized and delocalized hole components and their respective mobilities. Therefore, when the fraction of localized holes increases, the overall MSD growth rate decreases, manifesting as a reduction in the collective diffusion capability.

For Co_3O_4 , as the pump photon energy increases from 1.55 eV to 2.30 eV, the dominant transition pathway gradually shifts from MMCT to more localized transitions (including LMCT and possibly LF transitions). As a result, the proportion of oxygen holes within the total hole population increases significantly. To estimate this, we can perform multi-peak fitting of the steady-state absorption spectrum to separate the contributions from different transitions and approximate the proportion of delocalized holes at each excitation energy by calculating the relative fraction of MMCT-dominated absorption. Based on this approach, we estimate that the proportion of delocalized holes decreases from about 89.6% to 9.2%, leading to a marked reduction in the overall diffusion constant.

In principle, by combining the steady-state absorption spectrum (to estimate the relative weights of different transitions) with the MSD data under various pump wavelengths, one could attempt a partial deconvolution of the contributions from different excitation pathways. However, we note that such an analysis is highly dependent on the accuracy of the multi-peak Gaussian fitting model, while, as mentioned above, the precise spectral assignment of absorption bands above the band gap remains under debate. Therefore, using this model to quantitatively separate the diffusion behavior of individual holes may introduce significant model-dependent system uncertainties. For this reason, we did not apply spectral deconvolution or multi-component fitting to isolate the diffusion contribution of each transition. Instead, we focused on a qualitative analysis of the overall diffusion trends under different pump wavelengths and elucidating the underlying mechanisms.

Beyond the steady-state spectral estimates, we also attempted to differentiate the hole species based on differences in their excited-state lifetimes. TA measurements on Co_3O_4 under various pump wavelengths showed that the lifetime of the hole-induced absorption feature at 600 nm remains nearly unchanged across different excitation energies (**Figure R1.3**). This suggests that the lifetimes of metal and oxygen holes may be very similar, making it difficult to reliably separate their dynamical responses based on lifetime or signal amplitude alone.

Figure R1.3: The 600 nm PIA kinetics of Co_3O_4 at pump wavelengths of 340 nm, 530 nm and 700 nm.

Looking ahead, we believe that element-selective ultrafast probes, such as femtosecond O K-edge and Fe L-edge X-ray absorption spectroscopy (XAS), hold promise for directly tracking the dynamics of oxygen and metal holes, respectively. Such approaches could help to unambiguously resolve the relative contributions and dynamics of different hole species under various excitation conditions.

Action: In the wavelength-dependent transport measurement section of Co_3O_4 , we added: *“It should be noted that, due to the spectral overlap among different optical transitions, excitation at a given photon energy can simultaneously activate multiple excitation pathways. As a result, the measured diffusion constant reflects a population-averaged behavior of multiple hole species. As it is challenging to deconvolute contribution of different excitations, in this study, we focus on qualitatively correlating the observed diffusion trends with the dominant excitation characteristics under different pump wavelengths.”* We also included some discussions in the Supplementary Note 14.

4) *The question of whether highly mobile, band-like carriers are responsible for the primary photocurrents in hematite-based photoelectrochemical (PEC) devices remains unresolved. For instance, the authors cite work by Kay et al., which reports a quantum efficiency of only ~2% for carriers capable of long-range transport. Can the authors comment on the yield of these “hot” carriers in their experiments as a function of pump wavelength? Such an analysis would be valuable for assessing the potential device relevance of the observed fast transport regime.*

Response: We sincerely thank the reviewer for this insightful and important comment. To address the question of whether highly mobile hot carriers contribute significantly to the photocurrent in PEC devices, we conducted numerical simulations of carrier collection efficiency in hematite thin film photoelectrodes under varied excitation conditions. These simulations explicitly model the spatiotemporal evolution of both hot and cold carrier populations and consider factors such as wavelength-dependent optical absorption, carrier diffusivities, carrier recombination, and quantum yield of delocalized carriers.

Specifically, we simulated a 500 nm-thick hematite film using the wavelength-dependent transport parameters obtained in this work. The carrier extraction yield is calculated by integrating the net diffusive flux at the collecting interface over time. We define the early-time extraction yield as the fraction of photogenerated carriers collected within the first 2 ps following excitation, which serves as a proxy for the efficiency of ultrafast hot carrier collection. The total extraction yield refers to the cumulative fraction of carriers collected over the entire carrier lifetime, reflecting the overall collection efficiency.

Under 2.53 eV excitation, the relatively long photon penetration depth positions the initial carrier distribution closer to the collecting interface (**Figure R1.4 (a)**). However, due to the large contribution of localized transitions at this wavelength, we estimate that only ~61% of the photogenerated carriers are delocalized and thus capable of hot carrier transport. The resulting early-time and total extraction yields are 1.54% and 2.15%, respectively.

In contrast, under 3.02 eV excitation, despite a shorter penetration depth, the quantum yield of hot carrier generation is higher (~84%), and the associated diffusion coefficient is also larger

(Figure R1.4 (b)). As a result, the simulation predicts an early-time extraction yield of 8.07% and a total yield of 9.46%. These values are in good agreement with the enhanced photocurrent response observed under high photon energy excitation, as reported by Kay et al. These findings underscore the strong dependence of carrier transport and collection efficiency on pump wavelength and indicate a substantial contribution from hot carriers to the overall charge extraction process.

We note that the diffusion-based model employed here is idealized and does not account for several practical factors that may affect the quantitative extraction yield, such as drift currents induced by external bias, field-enhanced charge separation, or carrier losses due to interfacial recombination and bulk defects. In this model, hot and cold carrier populations are assigned fixed, wavelength-dependent diffusion constants over their respective timescales. This approach neglects the continuous, energy-dependent evolution of diffusivity during the carrier cooling process. Nonetheless, our results suggest that hot carriers make a non-negligible contribution to the photocurrent, particularly under high-energy photon excitation. The existence of a fast extraction channel associated with ultrafast hot carrier transport supports the potential relevance of hot carrier dynamics to device performance.

Figure R1.4: Simulated spatial distribution of photogenerated carriers in a 0.5 μm thick hematite film at different time delays under excitation with photon energies of (a) 2.53 eV and (b) 3.02 eV. The excitation surface is located at $x=0$, and the collecting interface is at $x=0.5 \mu\text{m}$.

Action: We have added the “To further evaluate their impact on the photocurrent response, we performed numerical simulations of carrier collection efficiency in a 500 nm-thick hematite”

film using the transport parameters...” at the end of result section in the manuscript. We have also added Supplementary Note 18 to describe the model and simulation results in detail.

Reviewer #2 (Remarks to the Author):

This work presents a description of carrier transport in two different TMOs (Co₃O₄ and α -Fe₂O₃) and decoupled the effect of hot hole transport and polaron hopping dynamics in these two materials. Femtosecond temporal and spatial resolution probing was utilized. The authors showed that these two materials have different behavior depending on the excitation energy and transition type, showing material specific carrier transport regimes. In my opinion, this work provides insightful interpretation of carrier and diffusion characteristics in TMOs. I recommend minor revisions before publication.

Response: We thank the reviewer for the positive and constructive feedback. In the revised manuscript, we have further clarified the excitation-dependent transport differences between Co₃O₄ and α -Fe₂O₃, justified the use of "nanoscopy," addressed the influence of film thickness, and added discussion and data on other TMOs.

1. The author mentioned that in case of α -Fe₂O₃, the diffusion constant is higher for LMCT-dominated excitation (3.02eV) than MMCT-dominated excitation (2.53eV), which is an opposite trend to Co₃O₄ case. In case of Co₃O₄, the diffusion constant is lower for LMCT-dominated excitation(2.68eV) than MMCT-dominated excitation(1.55eV). Can the author describe more explicitly why different TMOs have different diffusion constant behavior for the carrier generation mechanisms?

Response: We sincerely thank the reviewer for raising this insightful question. The opposite dependence of diffusion constants on excitation type in α -Fe₂O₃ and Co₃O₄ indicates the distinct role of optical transition characteristics and material-specific electronic structures in dictating hot carrier diffusion dynamics, beyond what has been generally believed that excess carrier energy solely dominates the hot charge diffusion dynamics.

While the underlying mechanisms remain not fully explored, we propose here one plausible scenario: the effect may be explained by the relative lifetimes of delocalized electronic states, governed by distinct localization pathways (e.g., polaron formation and/or trapping into ligand-field states) selectively activated by specific optical transitions, and to their intrinsic mobility

differences. The former defines the time window over which hot carriers maintain high mobility before self-trapping, whereas the latter dictates their diffusivity.

More specifically: α -Fe₂O₃ is a charge-transfer semiconductor with a valence band dominated by O 2p orbitals and strong on-site Hubbard repulsion (U) within Fe³⁺ 3d states (*Advanced Materials* 2018, 30, 1706577), whereas Co₃O₄ is a Mott–Hubbard semiconductor with substantial Co 3d character at the valence band edge and a comparatively smaller U. Recent transient extreme ultraviolet (XUV) spectroscopy has directly shown that the rate of polaron formation in metal oxides is strongly dictated by the nature of the optical excitation and its interplay with Hubbard U (*J. Am. Chem. Soc.* 2025, 147, 16018–16026). For instance, in GdFeO₃, MMCT excitation induces ultrafast (<250 fs) polaron formation, whereas LMCT excitation substantially suppresses localization. This contrast arises because different transitions perturb the local balance between Hubbard repulsion and super exchange interactions in opposite directions, either favoring or disfavoring localization. Therefore, the difference in hot-carrier transport observed in α -Fe₂O₃ and Co₃O₄ may reflect a fundamental divergence between charge-transfer and Mott–Hubbard electronic structures, mediated by the dynamics of polaron formation.

Furthermore, as suggested by the first reviewer, an additional contributing factor may involve the interaction between optical excitations and mid-gap localized ligand-field (LF) states. As reported by Sachs et al. recently (*Nat. Chem.* 2025, <https://doi.org/10.1038/s41557-025-01868-y>), open-shell TMOs such as α -Fe₂O₃ and Co₃O₄ host intrametallic LF states that serve as efficient sub-picosecond nonradiative decay channels. In Co₃O₄, LMCT excitations lie energetically close to these spin-allowed LF states, promoting ultrafast relaxation and strong carrier localization, thereby suppressing diffusivity. In contrast, MMCT transitions between Co²⁺ and Co³⁺ sites may partially bypass these LF-mediated localization, leading to enhanced charge transport. In α -Fe₂O₃, however, the LMCT→LF relaxation is spin-forbidden due to selection rules, requiring a spin flip before trapping can occur, thus prolonging the delocalized lifetime.

Taken together, these findings reinforce our central conclusion that the nature of the photoexcited transition, rather than the carrier energy alone, governs hot-carrier transport

behavior in transition metal oxides.

Action: We have revised the discussion section to provide deeper insights into the distinct transport mechanisms exhibited by different TMOs: *“While the precise mechanism remains unclear, we propose that this discrepancy may originate from the difference in the localization pathways following distinct optical transitions...., delaying localization and thereby extending the lifetime of delocalized transport.”*

2. In Figures 1a and 4a, the MMCT and LMCT appear in different colors, which is confusing.

Response and action: We thank the reviewer for pointing this out. To improve visual consistency and clarity of interpretation, we have revised the color scheme in Figure 4a to match that of Figure 1a.

3. In Figure 3b, can author describe more detail how the TA integration was obtained?

Response: We thank the reviewer for raising this important point. In Figure 3b, “TA integration” refers to the time-resolved integration of the TA signal over the 550-650 nm spectral window. This range captures the majority of the PIA feature centered near 600 nm, as shown in Figure 1b. Notably, this PIA band exhibits a pronounced spectral blue shift over time, which we attribute to the self-trapping of photogenerated holes. In Figure 3b, our analysis focuses on population-related spectral signatures, indicating that the observed decrease in terahertz photoconductivity arises primarily from reduced carrier mobility, rather than a decrease in carrier density.

To minimize artifacts introduced by spectral shifts, we integrated the TA signal over the full 550-650 nm window at each pump-probe delay. The resulting integrated signal represents the total population of photoexcited holes, including both free holes and localized species such as hole polarons, across the entire spectral evolution. Compared to monitoring a single probe wavelength, this method avoids distortions arising from spectral motion and provides a more reliable measure of the overall excited-state carrier population.

Action: We have revised the caption of Figure 3b for improved clarity: *“normalized integrated*

TA signal (dark blue dashed line) obtained by summing the PIA signal within 550-650 nm at each delay time, representing the total hole population including both free and localized species”

4. The authors describe their ultrafast optical measurement as "Nanoscopy". The optical setup in Figure S8 simply utilizing far-field optics whose resolution is set by diffraction limits. The author needs to justify the use of the term, or needs to use an alternative.

Response: We are grateful to the reviewer for putting forward this important point. Although our optical setup is based on far-field optics, the effective spatial accuracy for determining the diffusion length is neither determined by the size of the beam spot nor limited by the diffraction limit, but by the accuracy with which we can track the spatiotemporal evolution of the excited state population. Specifically, the diffusion constant is extracted from the temporal evolution of the Gaussian variance, and the uncertainty in the diffusion length is determined by the minimum detectable broadening of the spatial distribution over time. In this context, the spatial resolution is not constrained by the diffraction-limited spot size, but rather by the uncertainty in the Gaussian width extracted at different pump-probe delays, which is fundamentally limited by the signal-to-noise ratio of the system. We have now added a new section in Supplementary Note 9 to clarify this point in detail.

Figure R2.1: The raw data, fitting results and residuals of Co_3O_4 at 1.77 eV pump with representative time delays.

In particular, we selected three representative pump-probe delay times from our dataset to evaluate the uncertainty in spatial broadening. The corresponding raw data, two-dimensional Gaussian fitting results, and residual maps are now provided. Based on this analysis, the uncertainties in the extracted diffusion length at 0.3 ps and 100 ps are estimated to be ~18 nm and ~11 nm, respectively.

We estimate that our system achieves sub-20 nm spatial localization precision, allowing us to reliably track the nanoscale evolution of carrier distributions. This justifies our use of the term “nanoscopy” to describe the spatiotemporal probing of carrier dynamics with nanometer-level spatial precision.

Action: We have added new section in Supplementary Note 9 to further clarify the source of uncertainty in the extracted diffusion length in our ultrafast optical nanoscopy measurements.

5. How about other TMOs? Can the author provide an overall insight for the carrier dynamics and behavior of various other TMOs?

Response: We appreciate the reviewer's attention to the wide applicability of ultrafast transport of hot carriers in TMOs. In our study, we revealed the underlying mechanisms governing hot-carrier diffusion in two prototypical open d-shell TMOs, Fe_2O_3 and Co_3O_4 . Our results are in excellent agreement with recent findings (*Nat. Chem.* 2025, <https://doi.org/10.1038/s41557-025-01868-y>), which demonstrated that in open d-shell systems such as Fe_2O_3 , Co_3O_4 , Cr_2O_3 , and NiO, ligand-field (LF) states act as efficient sub-picosecond nonradiative localization channels that significantly limit spatial carrier transport.

In contrast, TMOs with d^0 or d^{10} configurations, such as TiO_2 , BiVO_4 , and CdO , inherently lack these LF-mediated pathways. In these materials, carrier localization is primarily governed by polaron formation dynamics. We note that ultrafast carrier motion has also been observed in d^0 TMOs such as BiVO_4 . A recent study on BiVO_4 (*J. Am. Chem. Soc.* 2024, 146, 31106-31113) reported spatially selective distributions of photogenerated electrons and holes within the first

few picoseconds following excitation. These observations suggest that non-equilibrium hot-carrier transport can also occur in closed d-shell (d^0) TMOs.

Figure R2.2: MSD evolution of Cu_2O at 2.53 eV and 3.1 eV excitations with delay time. The probe wavelength is 500 nm.

To further support our manuscript, we have included ultrafast optical nanoscopy measurements of Cu_2O (Figure R2.2), a representative d^{10} (closed d-shell) TMO, in the revised version. Remarkably, we observed similar hot-carrier diffusion dynamics in this system as well. These findings suggest that ultrafast hot carrier transport is not confined to binary open-shell TMOs, but may also be operative in multinary oxides with diverse d-electron configurations, including closed-shell systems.

Action: We have added a new paragraph in the discussion section: "Given the diversity of electronic structures across transition metal oxides, including variations in d-electron occupancy...., these findings suggest that ultrafast hot-carrier diffusion is likely a general phenomenon across a broad range of TMOs.". We have added Supplementary Note 19 containing ultrafast optical nanoscopy data for Cu_2O .

6. What is the TMO film thickness? Did the author account for the effect of film thickness (in terms of optical, carrier scattering effects)?

Response: We thank the reviewer for this important question. We have added cross-sectional

SEM images of the TMO films used in our study. The film thicknesses are approximately 60 nm for Co_3O_4 , 30 nm for $\alpha\text{-Fe}_2\text{O}_3$, and 220 nm for Cu_2O .

Figure R2.3 SEM cross-sections of the (a) Co_3O_4 , (b) $\alpha\text{-Fe}_2\text{O}_3$, and (c) Cu_2O thin films used in this study.

Our ultrafast measurements were performed in a transmission geometry, where the film thickness is significantly smaller than the optical penetration depth under our excitation conditions (typically several hundred nanometers in our cases). As a result, we can reasonably assume a uniform excitation profile across the film thickness, and the influence of optical field attenuation and axial absorption gradients can be neglected. This assumption simplifies the interpretation of the data and supports a diffusion model dominated by in-plane carrier transport.

To further validate this assumption, we performed optical nanoscopy measurements on a much thicker Co_3O_4 film ($\sim 50 \mu\text{m}$) under the same conditions as in Figure 2c. The extracted transport characteristics were consistent with those observed in the thin film, confirming that the reported ultrafast transport behavior is intrinsic to the material and excitation conditions, rather than being significantly affected by film thickness.

Figure R2.4 (a) SEM cross-sections of thick Co_3O_4 film (b) Evolution of MSD in the thick Co_3O_4 film as a function of pump-probe delay time. The pump wavelength is 700 nm and the probe wavelength is 570 nm.

Action: We have added the cross-sections SEM images in Supplementary Note 1. We have added the clarification “*(independent of film thickness, see Supplementary Note 6)*” after the sentence “*Two distinctive features are observed in the inferred time-dependent diffusion constant*” to clarify that the measured transport behavior is not affected by film thickness. In addition, we have included a new Supplementary Note 6, which provides detailed experimental evidence and discussion demonstrating the negligible influence of film thickness on the observed ultrafast transport dynamics.

Reviewer #3 (Remarks to the Author):

This manuscript presents a comprehensive study on ultrafast hot hole transport in transition metal oxides (TMOs), specifically Co_3O_4 and $\alpha\text{-Fe}_2\text{O}_3$, using ultrafast optical microscopy and terahertz spectroscopy. However, several key concerns about the interpretation of spectral features and dynamics must be addressed before publication.

Response: We thank the reviewer for the careful assessment of our work. In the revised manuscript, we provide experimental evidence for assigning the 600 nm feature to hole-related absorption, further clarify the attribution of polaron-associated spectral signatures, address the applicability of transient mobility extraction in THz measurements, and elucidate the excitation-dependent transport differences between Co_3O_4 and $\alpha\text{-Fe}_2\text{O}_3$.

1. The assignment of the 600 nm PIA to "photoinduced hole absorption from O 2p orbitals to the valence band maximum" (Lines 163-164) lacks direct experimental evidence. On the other hand, this feature has been assigned to the onsite d-d transition of the Co^{2+} ions in earlier works.

Response: We thank the reviewer for raising this important point. Regarding the assignment of the PIA feature at 600 nm, in this study we attribute it to photoinduced hole absorption based on the following considerations.

First, lifetime analysis (Fig. 1c) shows that if the 600 nm PIA originated from a Co^{2+} d-d excitation, which is a high-energy excited-state transition, it would be expected to decay faster than the CBM electron absorption at 910 nm. In contrast, our data reveals that the 600 nm signal persists significantly longer, suggesting an association with long-lived charge carriers rather than high-energy excited states.

Second, to provide direct evidence for the hole-related origin of the 600 nm feature, we have conducted scavenger experiments in the revision phase. In TMOs, methanol is a widely used hole scavenger that rapidly consumes photogenerated holes, whereas AgNO_3 serves as a standard electron scavenger. We measured the TA kinetics of Co_3O_4 at 600 nm in three media:

water, 10 % v/v MeOH, and 2 mM AgNO₃, while keeping the pump fluence constant. In the presence of MeOH, the 600 nm signal exhibits a markedly faster decay, consistent with the selective depletion of holes. In contrast, no change in the 600 nm dynamics is observed with AgNO₃, further confirming that the signal at this wavelength originates from holes.

Figure R3.1 TA kinetics of Co₃O₄ at 600 nm in water, with 10 % MeOH (hole scavenger), and with 2 mM AgNO₃ (electron scavenger).

Third, the decay timescale at 600 nm is on the order of hundreds of picoseconds to sub-nanoseconds and matches well with reported hole-polaron lifetimes in Co₃O₄ and other TMOs, supporting its assignment to a polaron state.

Finally, prior spectroelectrochemical studies (*J. Phys. Chem. C* 2014, 118, 3426-3432) have shown that a positive absorption near 600 nm is a clear signature of hole injection, whereas a bleach around 750 nm corresponds to electron population. This is consistent with our transient characteristics and assignment.

Taken together, we attribute the 600 nm PIA to photoinduced hole absorption from O 2p orbitals to states near the VBM. This feature can therefore serve as a reliable spectroscopic marker for hole self-trapping in Co₃O₄.

Action: We have clarified the relevant description in the revised manuscript in the *Assigning optical transitions in Co₃O₄* section: “To further identify the origin of this signal, we performed TA measurements on Co₃O₄ in the presence of selective electron and hole scavengers...from O 2p orbitals to the valence band maximum.” In addition, we have added Supplementary Note 3,

which provides scavenger experiment data and analysis.

2. In Fig. 1c, the kinetic trace probed at 600 nm shows distinct difference compared with those probed at 435nm and 910 nm. However, the difference kinetic trend cannot justify that the 600 nm PIA is associated with hole polarons. As shown in Fig. 1d, the d-d transition feature overlaps with the BGR induced positive feature, and this spectral overlap may account for a different kinetic trend at 600 nm.

Response: We thank the reviewer for the insightful comments on the kinetic characteristics of the 600 nm PIA. Here, we further clarify its assignment and address the possibility of spectral overlap.

Bandgap renormalization (BGR) typically arises from many-body Coulomb interaction effects at high carrier densities, where carrier-carrier interactions are modified under strong screening, leading to transient shifts of the conduction band and valence band edges. A characteristic signature of BGR is a pronounced pump-fluence dependence, with increasing carrier density leading to enhanced spectral broadening and energy shifts (*Nano Lett.* 2016, 16, 2945–2950; *Mater. Horiz.* 2023, 10, 4192–4201).

In our measurements, the TA spectra at 600 nm exhibits neither a noticeable change in the magnitude of the spectral shift nor any additional spectral broadening across different pump fluences (**Figure R3.2**). Given the characteristic fluence dependence of BGR, the absence of such a trend suggests that the 600 nm feature is unlikely to originate from BGR.

Figure R3.2 TA spectra of Co_3O_4 at 700 nm pump wavelength under different pump influences.

We also considered the possibility of thermally induced BGR, arising from lattice heating following carrier relaxation. Such processes are intrinsically slow, developing and decaying on nanosecond or longer timescales. By contrast, the dynamics we observe at 600 nm are significantly faster than those associated with lattice responses.

Taken together, these results indicate that neither BGR nor thermal effects can account for the 600 nm PIA. The absence of fluence dependence, coupled with the rapid spectral shift, is more consistent with photoinduced polaron absorption. We have clarified this point in the revised manuscript and noted in the Supplementary Information the possibility of spectral overlap and the reasons it can be excluded.

Action: We have revised Supplementary Note 4 to further discuss the potential influence of bandgap renormalization.

3. There may be an alternative interpretation for the apparent blue shift at 600 nm. As a consequence of the decay of BGR feature (PIA beyond 600 nm), the d-d transition centered at ~575 nm remains, giving rise to an apparent blue shift. Thus, there should be two different dynamics in this system: one is the sub-ps carrier localization due to the self-trapping. The other one is the recombination of the self-trapped excitons.

Response: We thank the reviewer for proposing an alternative interpretation of the apparent blue shift observed near the 600 nm PIA feature. As the reviewer points out that in complex transient absorption spectra, multiple spectral components may overlap or evolve over time, producing a dynamically changing observed spectrum. We therefore clarify the physical origin of the observed spectral shift.

If the blue shift were solely the result of two fixed spectral components—such as a BGR-related feature and a d-d transition—overlapping in time, then the decay of the BGR feature could indeed cause the apparent shift of the peak position towards shorter wavelengths. However, in such scenario, no rise in signal intensity at a new wavelength should be observed, as no new species would be generated. By contrast, our extracted kinetics at 570 nm and 615 nm clearly show that the 570 nm signal rises continuously within the first ps, accompanied by a decay at

615 nm, which is more consistent with a state-conversion process (**Figure R3.3**). As noted earlier, we attribute the PIA at 600 nm primarily to photoinduced hole absorption. Considering the energy stabilization associated with polaron formation, we regard it as more reasonable to assign this feature to the self-trapping of holes, accompanied by a change in transition energy that shifts the absorption center. This interpretation is further corroborated by our transport measurements.

Figure R3.3 TA dynamics of Co_3O_4 at different probe wavelengths.

Concerning the potential role of BGR decay, it is well established that the recovery of BGR is typically correlated with carrier recombination. If the observed blue shift is caused by the rapid decay of the BGR feature overlapping with the residual d-d transition, one would expect a rapid decrease in carrier density. However, our integrated signal over the ~550-650 nm range (Figure 3b) shows that the total carrier-associated absorption decreases only slowly, indicating a relatively stable carrier density and ruling out a significant decay of BGR as the main driver of the shift. This supports the conclusion that the observed shift reflects a genuine change in the absorption center rather than an artifact from spectral overlap.

Furthermore, the blue shift occurs within <1 ps after excitation, a timescale much faster than that typically associated with recombination of self-trapped carriers, which usually occurs over tens to hundreds of picoseconds (*J. Phys. Chem. Lett.* 2021, 12, 17, 4166–4171). Instead, prior studies have shown that polaron formation in TMOs normally occurs on the picosecond timescale (*J. Phys. Chem. Lett.* 2020, 11, 7867–7873), and such formation has been widely

reported to induce spectral shifts in TMOs (*J. Phys. Chem. Lett.* 2020, 11, 5686–5691). In summary, we conclude that the blue shift of the 600 nm PIA in Figure 1c is more reasonably attributed to the evolution of the absorption center during the ultrafast self-trapping of holes, rather than to BGR recovery or the persistence of a d-d transition.

Action: We have revised Supplementary Note 4 to provide a more comprehensive discussion on the possible origins of the blue shift, including the exclusion of BGR recovery and residual d-d transition overlap, and the justification for the self-trapping interpretation.

4. Given the fact of the “hot carriers” with sub-ps lifetime, the end of the THz pulse with a ps time cycle will experience a different response than the beginning of the pulse, which will complicate the analysis. Kindt and Schmittenmaer had carefully evaluated this issue in J. Chem. Phys. 1999, 110 (17), 8589-8596. The author should provide a detail procedure showing that the mobility of transient hot carriers can be extracted by the THz spectroscopy.

Response: We appreciate the reviewer’s insightful comment regarding the potential complications arising from the ultrafast decay of photoconductivity relative to the THz pulse duration. The reviewer is absolutely correct: in our case, the pump-induced photoconductivity changes faster than the duration of the THz pulse, requiring special care in tracking the pump-induced THz variations in the time domain.

Conventionally, THz time-domain spectroscopy (THz-TDS) studies have been performed by fixing the pump-probe delay while scanning only the delay stage of the sampling beam. This approach is valid when the carrier population has reached a quasi-steady state, such that the photoconductivity remains constant over the temporal width of the THz pulse. For rapidly decaying photoconductivity dynamics, extracting frequency-resolved photoconductivity (Figure 3d) by such measurement scheme becomes more challenging because the leading and trailing edges of the THz pulse interact with different carrier densities. To address this, we performed THz measurements with the sampling pulse (used for electro-optic detection) and the pump pulse (used for photoexcitation) held at a fixed temporal separation, while moving both delay stages simultaneously. This configuration ensures that all portions of the THz pulse

experience the same transient photoconductivity, i.e., the pump and sampling pulses are delayed together. Since the setup involves three beams (THz pulse, pump pulse, and sampling pulse), this detection scheme is equivalent to delaying only the THz generation pulse, thereby maintaining a constant pump-sampling delay for any given pump-THz probe delay.

Action: We have updated Supplementary Note 12 to address the applicability of THz spectroscopy for extracting the photoconductivity of hot carriers.

5. The sub-ps carrier localization shown by the THz kinetics is attributed to the hot carrier relaxation. However, the self-trapping of the photocarriers can give rise to the same decay dynamics. Can the self-trapping be excluded to account for the fast THz decay?

Response: We thank the reviewer for raising this important question and would like to take this opportunity to further clarify our assignment. In TMOs, the energy relaxation of hot carriers is governed by a competition between self-trapping, which leads to the formation of small polarons, and cooling via the emission of longitudinal optical (LO) phonons. As discussed in the main text, self-trapping often dominates, and the formation of localized polaron states with low mobility results in a pronounced reduction in photoconductivity.

To further investigate this, we compared normalized THz dynamics obtained at different pump photon energies (i.e., different excess energies; **Figure R3.4**). We found that the decay traces are nearly identical. This invariance with respect to excitation energy contrasts with the trend expected for a carrier-phonon interaction cooling process, in which higher excess energies should yield slower relaxation. Instead, the photon-energy-independent decay strongly supports a mechanism in which self-trapping precedes full carrier cooling. This interpretation is consistent with the temperature-dependent TA results of Zhang et al. (*J. Phys. Chem. Lett.* 2021, 12, 12033-12039), which likewise suggest that in Co_3O_4 , free carriers undergo self-trapping before sequential energy relaxation inside the bands.

Figure R3.4 Normalized photoconductivity dynamics of Co_3O_4 at different pump photon energies.

We note, however, that hot carrier cooling and self-trapping are difficult to fully disentangle within the sub-ps time window, and they may act cooperatively to produce the observed fast decay. We therefore attribute this process more generally to initial-state carrier localization, with self-trapping likely being the predominant driving force.

Action: We have added Supplementary Note 10 to provide a detailed discussion of the interplay between hot carrier cooling and self-trapping in Co_3O_4 .

6. This manuscript claims that the hot carrier diffusion coefficient depends on the pump photon energy (as shown in Fig.2d and Fig. 4b). Why the MSD for Fe_2O_3 and Co_3O_4 show different dependence on pump photon energy?

Response: We appreciate the reviewer's insightful questions regarding the opposite dependence of hot-carrier diffusivity on pump photon energy in $\alpha\text{-Fe}_2\text{O}_3$ and Co_3O_4 . We also note that Reviewer 2 raised a similar point (comment 1) and therefore provided a more detailed discussion here.

While excess carrier energy can influence the initial diffusion constant by increasing carrier kinetic energy, in TMOs the nature of the optical transition and the material-specific electronic structure play a more decisive role. We attribute the contrasting behavior of $\alpha\text{-Fe}_2\text{O}_3$ and Co_3O_4

to differences in ultrafast localization pathways and the intrinsic mobility of the electronic states populated by specific optical transitions. These ultrafast localization pathways critically define the ability of hot carriers to maintain high mobility before self-trapping occurs. Two key factors are considered:

1) Polaron formation dynamics:

In TMOs, the time window during which hot carriers retain high mobility is determined by the rate of localization, most prominently through small-polaron formation. The efficiency of this process is strongly influenced by how a given optical transition perturbs local electronic correlations. Optical excitation can alter the balance between on-site Hubbard repulsion (U) and super exchange interactions, thereby tipping the system towards either a localized or delocalized state.

α - Fe_2O_3 and Co_3O_4 exemplify two distinct scenarios. In α - Fe_2O_3 , the valence-band maximum is primarily derived from O 2p states, and the material behaves as a charge-transfer semiconductor with large U in Fe^{3+} 3d orbitals. By contrast, Co_3O_4 is a mixed-valent spinel whose valence-band edge has substantial Co 3d character, characteristic of a Mott-Hubbard system with comparatively smaller U . Recent transient extreme ultraviolet spectroscopy (*J. Am. Chem. Soc.* 2025, 147, 16018-16026) demonstrates that such differences in electronic structure profoundly affect localization rates, with the nature of optical excitation playing a central role. When the polaron formation time is shorter than the experimental resolution, carriers appear localized almost instantaneously, eliminating any measurable high-mobility regime. The opposite pump-energy dependence observed in α - Fe_2O_3 and Co_3O_4 therefore likely originates from fundamentally different polaron formation efficiencies, dictated by their distinct electronic structures and how these respond to specific optical transitions.

2) Coupling to ligand-field (LF) states:

Open-shell TMOs typically host low-energy localized LF states that can couple to photoexcited carriers and drive sub-picosecond localization (*Nat. Chem.* 2025, <https://doi.org/10.1038/s41557-025-01868-y>). The strength of this coupling strongly depends on the material's electronic structure, the nature of the optical transition, and the spin selection

rules. In Co_3O_4 , LMCT-excited carriers couple strongly and spin-allowably to LF states, promoting ultrafast trapping and suppressing diffusivity. In $\alpha\text{-Fe}_2\text{O}_3$, however, LMCT to LF relaxation is spin-forbidden, reducing coupling efficiency and thereby allowing more delocalized transport under LMCT excitation.

Furthermore, recent computational studies (*Advanced Science* 2024, 11, 2306243) have demonstrated that carrier localization in TMOs is highly element-selective and orbital-specific. In Al_2O_3 , for instance, electron localization is governed by the local cation coordination and the orbital character of neighboring atoms. Such sensitivities likely contribute to the material-dependent localization efficiencies observed here.

In summary, while the pump photon energy sets a fundamental parameter for tuning the initial kinetic energy of hot carriers, the ability to sustain long-range transport is ultimately constrained by a combination of factors, including local orbital configurations, the distribution of ligand-field states, spin selection rules, and the excitation pathway. The contrasting wavelength-dependent diffusivity trends observed in $\alpha\text{-Fe}_2\text{O}_3$ and Co_3O_4 therefore reflect a deeper coupling between hot-carrier dynamics and the underlying electronic structure of the material.

Action: We have revised the discussion section to provide deeper insights into the distinct transport mechanisms exhibited by different TMOs: “While the precise mechanism remains unclear, we propose that this discrepancy may originate from the difference in the localization pathways following distinct optical transitions....., delaying localization and thereby extending the lifetime of delocalized transport.”